# Locus-Specific Enrichment Analysis of 5-Hydroxymethylcytosine Reveals Novel Genes Associated with Breast Carcinogenesis

**DOI:** 10.3390/cells11192939

**Published:** 2022-09-20

**Authors:** Deepa Ramasamy, Arunagiri Kuha Deva Magendhra Rao, Meenakumari Balaiah, Arvinden Vittal Rangan, Shirley Sundersingh, Sridevi Veluswami, Rajkumar Thangarajan, Samson Mani

**Affiliations:** 1Department of Molecular Oncology, Cancer Institute (WIA), 38, Sardar Patel Road, Chennai 600036, Tamilnadu, India; 2Department of Oncopathology, Cancer Institute (WIA), 38, Sardar Patel Road, Chennai 600036, Tamilnadu, India; 3Department of Surgical Oncology, Cancer Institute (WIA), 38, Sardar Patel Road, Chennai 600036, Tamilnadu, India

**Keywords:** DNA methylation or 5-methylcytosine, 5-hydroxymethyl cytosine, methylation imbalance, breast cancer, DMR, DhMR, enrichment analysis

## Abstract

**Highlights:**

**Abstract:**

An imbalance in DNA methylation is a hallmark epigenetic alteration in cancer. The conversion of 5-methylcytosine (5-mC) to 5-hydroxymethyl cytosine (5-hmC), which causes the imbalance, results in aberrant gene expression. The precise functional role of 5-hydroxymethylcytosine in breast cancer remains elusive. In this study, we describe the landscape of 5-mC and 5-hmC and their association with breast cancer development. We found a distinguishable global loss of 5-hmC in the localized and invasive types of breast cancer that strongly correlate with *TET* expression. Genome-wide analysis revealed a unique 5-mC and 5-hmC signature in breast cancer. The differentially methylated regions (DMRs) were primarily concentrated in the proximal regulatory regions such as the promoters and UTRs, while the differentially hydroxymethylated regions (DhMRs) were densely packed in the distal regulatory regions, such as the intergenic regions (>−5 kb from TSSs). Our results indicate 4809 DMRs and 4841 DhMRs associated with breast cancer. Validation of nine 5-hmC enriched loci in a distinct set of breast cancer and normal samples positively correlated with their corresponding gene expression. The novel 5-hmC candidates such as TXNL1, and CNIH3 implicate a pro-oncogenic role in breast cancer. Overall, these results provide new insights into the loci-specific accumulation of 5-mC and 5-hmC, which are aberrantly methylated and demethylated in breast cancer.

## 1. Introduction

DNA methylation imbalance is one of the hallmark epigenetic events in cancer. Cytosine DNA methylation (5-methylcytosine or 5-mC) occurs at the gene promoter and is often associated with gene repression, while its oxidized form, 5-hydroxymethylcytosine (5-hmC), relaxes the repression [1,2]. The oxidation of 5-mC to 5-hmC is catalyzed by the *TET* family of genes *TET1*, *TET2*, and *TET3* [3,4,5,6,7]. Based on tissue specificity, 5-hmC levels can vary between 0.1% and 1% of the human genome [8]. The increase in 5-hmC is strongly associated with transcriptional activation [9]. Effective binding of methylation readers such as *MBD3* and *MeCP2* preferentially to 5-hmC results in active transcriptional assembly and activity [10,11]. It turned out that 5-hmC is the stable epigenetic modification involved in the transcription machinery, and does not just serve as an intermediate in the demethylation process. The imbalance between 5-mC and 5-hmC is of recent interest as both are associated with gene expression and lead to carcinogenesis.

A global reduction of 5-hmC is evident in several cancers [8,12,13,14,15]. Studies on melanoma, pancreatic cancer, lung cancer, and prostate cancer suggest that aberrant 5-mC and 5-hmC levels may predispose to tumor progression [16,17,18,19,20,21,22,23]. However, there is limited evidence for 5-mC and 5-hmC dynamics in breast cancer. We hypothesize an imbalance among the genomic 5-mC and 5-hmC levels that contribute to breast carcinogenesis. Previous reports affirm that 5-hmC levels depend on tissue-specific *TET* expression. Particularly, downregulation of the genes *TET1* and *TET2* have been reported to alter the 5-hmC levels [24]. A recent study on breast cancer showed the altered 5-hmC profiles and their association with lymph node metastases [25]. In breast cancer, the locus-specific deposition of 5-hmC and its functional role in the control of gene expression are poorly understood. Emerging enrichment approaches can identify 5-mC and 5-hmC genomic regions with single-base resolution and describe the differentially methylated regions (DMRs) and differentially hydroxymethylated regions (DhMRs) in cancer [26,27]. Determining the 5-hmC modified genomic regions in breast cancer will be useful for diagnostic and therapeutic markers.

In this study, we found that global methylation and hydroxymethylation levels were drastically reduced in breast cancer tissues. The global 5-hmC reduction was associated with the downregulation of the *TET1* and *TET3* genes. The genome-wide analysis revealed differentially methylated and differentially hydroxymethylated breast cancer loci. We also identified a strong correlation between the 5-hmC alterations and gene expression changes. Altogether, the study provides a comprehensive genome-wide distribution of 5-mC and 5-hmC, and also an imbalance in the DNA methylation machinery that leads to breast cancer development.

## 2. Materials and Methods

### 2.1. Clinical Specimen

Breast cancer tissues of stages IIA–IV and paired normal (PN) tissue were obtained from the Tumor Bank, Cancer Institute (WIA), Chennai, India. Tissue samples were collected from patients undergoing direct surgery for invasive ductal carcinoma (IDC) or ductal carcinoma in situ (DCIS). Tumor tissues (*n* = 15) were histopathologically confirmed to consist of >70% tumor cells, and paired non-cancerous tissue (*n* = 15) free of tumor cells was excised away from the tumor margin. Similarly, DCIS (*n* = 5) were obtained from patients undergoing a wide-excision biopsy, and absolute normal samples (*n* = 5) were collected from patients undergoing wide-excision biopsy for non-tumorous conditions, such as fibrosis or adenosis, and histopathologically confirmed to be free of any tumor cells. An additional set of tumour samples (*n* = 30) and non-cancerous (*n* = 6) tissues were also collected for a validation study. Informed consent for participation and sampling was obtained from all patients. The Cancer Institute (WIA), Institutional Ethics Committee (IEC/2016/05) approved the study.

### 2.2. Isolation of Genomic DNA

About 25 mg of tissue was homogenized and DNA was isolated using the Nucleospin Tissue DNA Kit (Macherey Nagel, GmbH, Düren, Germany) according to the manufacturer’s instructions. The isolated DNA was quantified with Nanodrop ND-2000 and stored at −20 °C until further use.

### 2.3. Estimation of Global Levels of 5-hmC and 5-mC

Genomic DNA (100 ng) was used for the estimation of global 5-hmC and 5-mC levels by ELISA using the Quest 5-hmC ELISA kit and the 5-mC DNA ELISA kit (Zymo Research Inc., Irvine, CA, USA) according to the manufacturer’s instructions.

### 2.4. RNA Isolation and TET Expression Assay

Briefly, tissues were homogenized, and RNA was isolated using the Nucleospin^®^ RNA Isolation Kit (Macherey Nagel, GmbH). RNA was quantitated using Nanodrop ND-2000 and cDNA was synthesized from 500 ng of total RNA using a Quantitect^®^ reverse transcription kit (Qiagen, Germantown, MD, USA). Gene expression analysis of *TET* 1, 2, and 3 was performed using TaqMan probes (Appendix A), TaqMan™ Universal Master Mix II, no UNG (Applied Biosystems, Waltham, MA, USA), and the Quant studio 12Kflex system (Applied Biosystems, Waltham, MA, USA).

### 2.5. Enrichment of 5-mC Modified DNA Regions and Library Preparation

Genomic DNA (1 μg) was fragmented to 300–600 bp after 25 cycles of 30 s of pulsed sonication in Bioruptor (Diagenode, Liège, Belgium). Fragmented DNA was end-repaired, and adapter ligation was performed using the NEBNext Ultra II DNA Library Prep Kit (New England Biolabs, Ipswich, MA, USA) for Illumina. Furthermore, fragments with a length of 400–500 bp (300 bp insert + 120 bp adapter) were size-selected using AMPure beads (Beckman Coulter, Brea, CA, USA). Immunoprecipitation of methylated DNA was performed using the MagMeDIP kit (Diagenode, Belgium), and enriched DNA was purified using the I Pure kit (Diagenode, Belgium) according to the manufacturer’s protocol. Purified DNA was indexed and amplified using the NEBNext Ultra II DNA Library Prep Kit (New England Biolabs, Ipswich, MA, USA).

### 2.6. Enrichment of 5-hmC Modified DNA Regions and Library Preparation

For the enrichment of 5-hmC-modified DNA, a reduced representation hydroxymethylation profiling by RRHP kit (Zymo Research Inc., Irvine, CA, USA) was carried out. The genomic DNA (1 μg) was digested using the *Msp1* enzyme and ligated with p5 and p7 adapters. The adapter-ligated fragments were glycosylated with UDP-glucose and T4-glycosyltransferase and digested again with *Msp1* to cleave adapters from non-glycosylated fragments. The fragments were size selected 400–500 bp (300 bp insert + 120 bp adapter) using AMPure XP beads (Beckman Coulter, Brea, CA, USA) and amplified as libraries using RRHP™ 5-hmC Library Prep Kit (Zymo Research. Inc., Irvine, CA, USA).

### 2.7. Sequencing and Data Analysis

Enriched libraries were sequenced by 150 × 2 paired ends to generate 25 million reads in the Illumina Nextseq 500. The FASTQ files were quality checked by FastQC. Trimmomatic was used to trim the adapter sequence and the low-quality reads (Phred > 30) were discarded. Using the BWA mem algorithm, the processed fastq files were then aligned to the reference genome (hg19). Aligned files were then converted into sorted bam files using samtools. The peaks were called with MACS2, and the peak files were used to find overlapping peaks in multiple files, and their raw counts were extracted with the DiffBind R package.

### 2.8. DMR Analysis

DMR analyses were carried out using the DESeq2 R package and the likelihood ratio test (LRT) was used to determine DMR with the padj value cut-off of ≤0.01. Furthermore, the DMR regions were filtered based on the number of peaks called in biological replicate with at least 10 | 10 | 5 | 5 (T | PN | DCIS | AN). Filtered DMR was annotated using the ChIPseeker R package with the UCSC hg19 known gene sets. The Z-Score was calculated from normalized counts and the heatmap was plotted using the Complex Heat map R package. The sorted bam files were indexed, and the coverage profile was calculated using the Deep Tools bam Coverage with RPKM normalization and a bin size of 20 bp. The resulted bigwig files were used to visualize peak regions with IGV.

### 2.9. DhMR Analysis

For the DhMR analysis, RPKM values were extracted from the DiffBind R package and a global rank-invariant set normalization was carried out using the rank-invariant function from the Lumi R package. The Kruskal–Wallis test was performed, and the *p*-value was adjusted with FDR. A cut-off of padj ≤ 0.1 was set to define DhMR. In addition, the DhMR peaks were filtered with the same criteria as DMR. The chromosomal distributions of DMR and DhMR were analyzed with the Karyoplot R-Package with a *p*-value of 0.05. The ideogram was examined for the DMR and DhMR regions with an FDR of 0.05.

### 2.10. Pathway Enrichment and Gene Ontology Analysis

Pathway enrichment and gene ontology analysis were performed with the g: Profiler and Cluster Profiler R package. For the *p*-adjusted corrected method, FDR with a *p*-value cut-off of 0.05 was used. Dot blots were generated based on the criteria mentioned above both for DMRs and DhMRs.

### 2.11. Validation of Loci-Specific 5-hmC Enriched Regions Using qPCR Assays

Briefly, the tumor (*n* = 30) and normal tissue (*n* = 6) were homogenized, and DNA was isolated using the Nucleospin Tissue DNA Kit (Macherey Nagel, GmBH) according to the manufacturer’s instructions. The genomic DNA was subjected to a 5-hmC-specific enrichment with EpiMark 5-hmC and 5-mC analysis kit (New England Biolabs, Ipswich, MA, USA). The processed DNA samples were analyzed with qPCR using loci-specific primers (Appendix A). Gene expression, methylation analysis, and survival analysis were carried out with the UALCAN [28] web tool for 5-mC- and 5-hmC-specific candidate genes.

### 2.12. Statistical Analysis

The correlation analysis was performed between the expression levels of *TET* enzymes and the global levels of 5-mC and 5-hmC distribution using GraphPad Prism v 7.0a (GraphPad Software, La Jolla, CA, USA). Spearman’s rank correlation test was performed with a confidence interval of 95% for generating the correlation plot. Linear regression lines were generated based on the R-value of the entities. The Wilcoxon signed-rank test was used for the nonparametric paired analysis. The Mann–Whitney U test was performed for the nonparametric unpaired analysis. The *p*-value of <0.05 is considered a significant outcome.

## 3. Results

### 3.1. Loss of 5-hmC Is Associated with TET 1 and TET3 Downregulation in Breast Cancer

Global levels of 5-hmC and 5-mC were first quantified in breast cancer (invasive ductal carcinoma (IDC) and ductal carcinoma in situ (DCIS)) paired normal (PN), and apparent normal (AN) tissues. We found a significant decrease in 5-hmC levels in IDC vs. PN (FC = −2.58, *p* = 0.0003) and DCIS vs. AN (FC = −2.08, *p* = 0.0324) (Figure 1a). Although global 5-mC levels were reduced in IDC vs. PN (FC = −2.28, *p* = 0.0014), there was no significant difference between the DCIS vs. AN group (*p* = 0.5467) (Figure 1b). The results imply that the global loss of 5-hmC is a characteristic epigenetic alteration of localized and invasive breast cancer. The differential expression of *TET* genes leads to the altered 5-hmC levels in breast cancer. Therefore, we quantified the *TET* gene expression levels and found that the genes *TET2* (FC = +2.02, *p* = 0.0317) (Figure 1d), and *TET3* (FC = +2.0, *p* = 0.0159) (Figure 1e) were upregulated in DCIS vs. AN group. However, *TET1* (FC = −2.08, *p* = 0.0266) (Figure 1c) and *TET3* (FC = −2.0, *p* = 0.026) (Figure 1e) genes were downregulated in the IDC vs. PN and IDC vs. AN groups. Spearman’s rank test showed no significant correlation of *TET* genes with global 5-hmC levels in PN tissues (Figure 1f–h), while it showed a significant positive correlation of *TET1* (*r* = 0.544, *p* = 0.05) (Figure 1i) and *TET3* (*r* = 0.5662, *p* = 0.0437) (Figure 1k) genes with the global loss of 5-hmC in breast cancer but not with the *TET2* gene (Figure 1j). In addition to *TET1*, we report here that *TET3* is also associated with a global loss of 5-hmC in breast cancer.

### 3.2. Relative Abundance of 5-hmC in Breast Cancer and Their Enrichment at Distal Regulatory Sites

We performed the enrichment of genomic 5-mC and 5-hmC specific regions followed by high-throughput sequencing and achieved ~14 million reads from 5-hmC-enriched libraries and ~21 million reads from 5-mC-enriched libraries. Almost 99.7% of the 5-hmC reads and ~98.94% of 5-mC reads were mapped effectively against the reference genome. Principal component analysis (PCA) and hierarchical clustering analysis (HCA) showed a clear pattern of segregation from the tumor (IDC and DCIS) to normal samples (PN and AN) (Appendix A). MACS-2 peak calling of the mapped reads resulted in a total of 3.3 million 5-hmC-enriched peak sets and 4.7 million 5-mC-enriched peak sets. Differential peak calling analysis between breast cancer and normal groups identified 4809 differential methylated regions (DMRs) (*p* < 0.01, FDR < 0.05) (Figure 2a) and 4841 differential hydroxymethylated regions (DhMRs) (*p* < 0.01, FDR < 0.05) (Figure 2b) (Appendix A). The distribution of peaks across the chromosomes showed a higher peak intensity of DhMR over DMR (Figure 2c,d) (window size: 1 × 10^−6^). Significantly higher peak intensity of DhMR indicates a potential difference in loci-specific 5-hmC levels between breast tumor and paired normal samples.

The peaks characterized by their genomic features indicated that both DMR and DhMR were moderately found in the gene body (28.97% and 37.31%, respectively), particularly in the intronic regions, but not in the exons. The DMR profile was high in the promoter (28.57%) regions. However, only a mere 8.82% enrichment of DhMR was found in the promoter region compared to massive accumulation in the distal intergenic regions (43.11%) ((Figure 2e) (Appendix A)). Therefore, we show here that the distribution in the gene body does not invariably differ between the two modifications, but DMR is typically enriched at the proximal regulatory sites (promoter), while DhMR at the distal regulatory sites (intergenic region). We found the enrichment of 5-mC and 5-hmC were significantly distinguishable between tumor and normal tissues. Further, the enriched peak sets of the DhMRs and DMRs were analyzed for transcription factor binding sites. We found that the accumulation in the promoter regions of DMRs in the interval 1500 bp upstream and 1500 bp downstream of the TSSs (read count frequency > 6.5 × 10^−5^) was higher than that of DhMRs (read count frequency < 3 × 10^−5^) (Appendix A). While in DhMRs, the accumulation was observed in the distal intergenic regions upstream 5 kb from the TSSs (read count frequency > 6.5 × 10^−5^) ((Figure 2f–g) (Appendix A)). We also used LOLA-Web to identify the locus overlap between the 5-hmC sites and the regulatory sites in the distal intergenic regions (i.e., regions > −5 kb from the TSSs). Tumor and PN peak sets of 5-hmC were tested against the ENCODE data set with the reference genome hg19 and normalized with preloaded Tiles1000.hg19.bed. Tumor-specific 5-hmC peak sets are significantly associated with the enhancer sites of the breast cancer cell line MCF-7 regions such as H3K4me1, H3K4me3, H3K14ac, and H3K9ac (log (*p*-value) > 300). Enhancer sites overlapped with loci-specific 5-hmC levels of the candidate genes in the breast cancer cell lines, such as MDA-MD-468, MDA-MB-231, and MCF-7 regions, but not in the normal luminal cell line MCF-10A (Appendix A). Hence, the results suggest that DhMRs were widespread in the distal regulatory regions, while DMRs accumulated in the proximal regulatory regions of the breast cancer genome.

### 3.3. Locus-Specific Imbalance of DhMRs and DMRs in Breast Cancer

To determine the exact loci and the differential distribution of 5-hmC and 5-mC accumulation in breast cancer, we identified 35 hyper-hmC loci (Appendix A), and 30 hypo-hmC loci (Appendix A). The hyper-hmC loci included coding genes (*GALC*, *BNIPL*, *TXNL1*, *CNIH3*, etc.), lncRNA (*LINC00535*, *LINC00662*, and *PTPRN2* lncRNA), and microRNA (*MIR4278*, *MIR1204*, *MIR944*, and *MIR921*). We found 26 coding genes (*ZBTB16*, *SP8*, *THRB*, *HIC2*, etc.,) and only four non-coding loci (*MIR4417*, *MIR3612*, *LINC00911*, and *LINC00417*) among the hypo-hmC-specific regions. A total of 57 hyper-mC loci inclusive of 53 coding genes (*CCDC181*, *SIM2*, *ID4*, etc.) and 4 non-coding genes (*LINC01257*, *LOC728989*, *MIR5087*, and *MIR183*) were obtained. The hypo-mC loci consisted of 24 coding genes (*OPCML*, *MKI67*, and *SPOCK1*) and 6 were non-coding (*MIR548AR*, *LINC02347*, *MIR744*, *MIR3612*, etc.) (Figure 3a) (Appendix A). Further, we validated five hyper-hmC and four hypo-hmC loci in breast cancer and normal tissues. The results confirmed the 5-hmC gain of *TXNL1* (FC = 4, *p* = 0.0102) (Figure 3b,c), *CNIH3* (FC = 2, *p* = 0.0242) (Figure 3d), while for *BNIPL*, *A4GALT* and *CBLN4* no statistical significance was observed, although they followed the same trend (Figure 3e–g). On the other hand, hypo-hmC candidates *CHODL* showed a two-fold loss of 5-hmC (*p =* 0.0416) (Figure 3h). Other loci such as *ZBTB16*, *HIC2,* and *SP8* showed a trend towards 5-hmC loss in tumors but did not show any statistical significance (Figure 3i–k). Our validation analysis confirmed that the gain of 5-hmC in *TXNL1*, *BNIPL*, *CNIH3* and loss of 5-hmC in *CHODL*, *ZBTB16*, *SP8*, and *HIC2* in breast cancer samples. The gene function prediction by g: Profiler indicated that 5-mC and 5-hmC genes were related to cell cycle regulation, cell cycle inhibition, and cell proliferative signals that could potentially affect breast cancer development and progression (Appendix A). The gene ontology and KEGG pathway enrichment map of DhMRs and DMRs also revealed the association of breast cancer-specific 5-hmC and 5-mC with transcriptional machinery (Appendix A). Altered levels of 5-mC and 5-hmC eventually control and determine the progression of breast cancer development. Extensive functional analysis of the identified loci will elucidate the significance of the 5-mC and 5-hmC imbalance in breast cancer development.

### 3.4. Association of 5-mC and 5-hmC Modifications with Gene Expression

We investigated the aberrant levels of methylation and hydroxymethylation in the locus-specific aspect and its impact on the regulation of gene expression using the TCGA-breast cancer data set (UALCAN webtool) (Table 1). We found the hyper-methylation of *CCDC181* (beta value > 0.5, *p* < 0.05), *ID4* (beta value > 0.5, *p* < 0.05) and hypo-methylation of *MKI67*, *OPCML*, and *SPOCK1* (beta value < 0.3, *p* < 0.05). Correspondingly, *CCDC181* (normalized TPM count = −0.094, *p* < 0.1) (Figure 4a), *OR4F29* (normalized TPM count = −0.005, *p* < 0.1) (Figure 4b), and *ID4* (normalized TPM count = −58.839, *p* < 0.001) (Figure 4c), were downregulated in tumor samples (*n* = 1094) while, *MKI67* (normalized TPM count = 9.94, *p* < 0.001) and *SPOCK1* (normalized TPM count = 3.26, *p* < 0.001) were found to be over-expressed in tumor samples but not *OPCML* (Figure 4d–f). The inverse correlation confirmed that hypermethylation leads to the suppression of gene expression and hypomethylation leads to overexpression of the gene. Further, we found that hyper-hmC candidates, *BNIPL* (normalized TPM count = 5.344, *p* < 0.05), *CNIH3* (normalized TPM count = 0.778, *p* < 0.005), and *TXNL1* (normalized TPM count = 14.549, *p* < 0.001) to be upregulated and the hypo-hmC candidates such as *ZBTB16* (normalized TPM count = −13.933, *p* < 0.001), *HIC2* (normalized TPM count = −0.436, *p* < 0.001), *CHODL* (normalized TPM count = −1.093, *p* < 0.001), *THRB* (normalized TPM count = −13.792, *p* < 0.001) and *RAPGEF2* (normalized TPM count = −9.296, *p* < 0.001) were significantly downregulated (Figure 4g–p). Several loci relax the repressive methylation marks by increasing the 5-hmC levels leading to gene activation. In this study, we found that three candidate genes, *TXNL1*, *CNIH3*, and *BNIPL*, showed an increase in 5-hmC levels associated with gene overexpression. The direct proportionality between 5-hmC and gene expression reinstates that gain of 5-hmC activates gene transcription. We further investigated gene expression influence and the candidate gene’s overall survival with hyper-hmC specifications. The results indicate that overexpression of genes such as *TXNL1* (*p* = 0.042) (Appendix A), *CNIH3* (*p* = 0.26) (Appendix A) and *BNIPL* (*p* = 0.0001) (Appendix A) was associated with poor overall survival in breast cancer patients.

## 4. Discussion

The present study illustrated the methylation and hydroxymethylation landscape of the breast cancer genome. Initial findings showed that the downregulation of the *TET* genes causes a global 5-hmC reduction in breast cancer. The genome-wide profiling revealed a higher 5-mC accumulation around the TSSs from −1.5 k to +1.5 k, while the 5-hmC accumulated in the intergenic regions (>−5 kb away from TSSs). Thus, DMRs are mainly associated with proximal gene regulation and DhMRs with distal regulation. We found an intergenic and gene body gain of 5-hmC associated with gene overexpression and a loss of 5-hmC towards downregulation of the corresponding genes. The study results show that the imbalance between 5-mC and 5-hmC is a novel phenomenon orchestrating the epigenetic machinery of breast cancer.

Previous studies also reported global 5-mC and 5-hmC loss in various cancers, including breast cancer [16,22,29,30]. The tissue-specificity and the alterations of *TET* gene expression in advancing cancer stages determine the 5-hmC levels and the demethylation process [13,17,23,31]. The cytoplasmic mislocalization of *TET1* in ER/PR-negative subtypes of IDC and DCIS was directly proportional to the global reduction in 5-hmC levels [24]. We show here that the global reduction in the 5-hmC content in IDC is dependent on *TET1* and *TET3* genes. The CXXC domains of *TET1* and *TET3* enhance the DNA binding efficiency at DNA demethylation sites [32]. The expression and nuclear import of *TET1* and *TET3* facilitate an active oxidation process. Our finding shows the downregulation of *TET3* also contributes to 5hmC loss in breast cancer tissues.

Genome-wide profiling showed the effects of promoter methylation in breast cancer. The advancement in enrichment strategies and next-generation sequencing has resulted in the contemporary analysis of 5-mC and 5-hmC describing their genomic signatures in breast cancer [33,34,35]. In our study, we found the enrichment of 5-mC and 5-hmC significantly distinguishable between tumor and normal tissues. Higher hydroxymethylation levels were observed in the promoter and UTRs of DCIS tissues. On the other hand, the invasive type showed a higher accumulation in the gene body and intergenic regions than in the promoter regions. Hence, we speculate that accumulation of 5-hmC in the preliminary stage of breast cancer occurs mainly at the proximal regulatory regions, reducing the suppression caused by 5-mC. In the locally advanced breast tumors, the enrichment at the distal intergenic regions implicates an enhancer-like activity of 5-hmC. Previous studies also reported that 5-hmC could potentially act as an enhancer or super-enhancer elements ~5–10 kb and >20 kb away from the TSSs [36,37]. The overlapping histone markers from the ENCODE roadmap project and 5-hmC sites emphasized the association of active enhancer sites in the breast cancer genome. The positive association of histone activation and 5-hmC gain suggests a synergistically enhanced gene.

Several loci relax the repressive methylation marks by increasing the 5-hmC levels leading to gene activation. Our results found *TXNL1* as a novel 5-hmC candidate gene in breast cancer with an increased 5-hmC level and corresponding gene overexpression. Survival analysis also indicated the overexpression of *TXNL1* and *BNIPL* in breast cancer is significantly associated with poor overall survival. Previous studies reported that induction of oxidative stress led to the overexpression of *TXNL1* associated with the downregulation of the DNA repair protein XRCC1, an accumulation of DNA damage, and *BCL-2* regulation [38]. Further functional analysis on the 5-hmC gain at *TXNL1* and other loci would be warranted to elucidate the mechanistic insight of 5-hmC in the transcriptional activation and breast cancer development.

## 5. Conclusions

The present study opens a new paradigm on the imbalance of 5-mC and 5-hmC in breast cancer. The study offers a detailed perspective on an epigenomic instability substantiated by the loss of 5-mC and 5-hmC in breast tumors. Global loss of 5-hmC is associated with *TET1* and *TET3* downregulation. Genome-wide profiling has revealed a profound imbalance in breast cancer’s region-specific distribution of 5-mC and 5-hmC. Predominant 5-hmC modifications localized at distal gene regulatory sites implicating a transcription enhancing function. The novel 5-hmC candidates identified in the study can be promising diagnostic and therapeutic markers for breast cancer.

## Figures and Tables

**Figure 1 cells-11-02939-f001:**
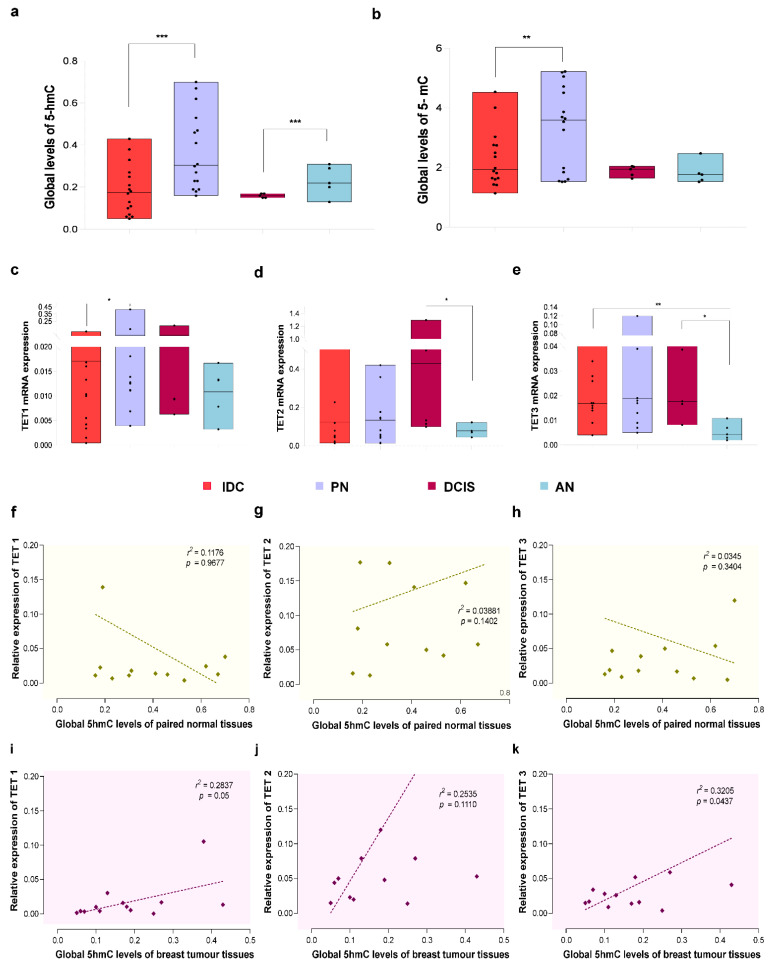
Global 5-mC, 5-hmC, and *TET* gene expression levels in breast cancer. (**a**) Global levels of 5-hmC in IDC (*n* = 15), PN (*n* = 15), DCIS (*n* = 5), and AN (*n* = 5) tissues. (**b**) Global levels of 5-mC in IDC (*n* = 15), PN (*n* = 15), DCIS (*n* = 5), and AN (*n* = 5) tissues. (**c**) Gene expression analysis of *TET1* among the IDC, PN, AN, and DCIS samples. (**d**) Gene expression analysis of *TET2* among the IDC, PN, AN, and DCIS samples (**e**) Gene expression analysis of *TET3* among the IDC, PN, AN, and DCIS samples. (**f**) Correlation analysis of global 5-hmC levels of PN samples with relative mRNA expression of *TET1* gene. (**g**) Correlation analysis of global 5-hmC levels of PN samples with relative mRNA expression of *TET2* gene. (**h**) Correlation analysis of global 5-hmC levels of PN samples with relative mRNA expression of *TET3* gene. (**i**) Correlation analysis of global 5-hmC levels of breast tumour samples with relative mRNA expression of *TET1* gene. (**j**) Correlation analysis of global 5-hmC levels of breast tumour samples with relative mRNA expression of *TET2* gene. (**k**) Correlation analysis of global 5-hmC levels of breast tumour samples with relative mRNA expression of *TET3* gene. The Wilcoxon signed-rank test was applied to test the statistical significance of paired analysis. The Mann–Whitney U test was used to evaluate unpaired or grouped analysis (*** *p* < 0.0001; ** *p* < 0.001, and * *p* < 0.05). Abbreviations: PN = paired normal; IDC = invasive ductal carcinoma; AN = apparent normal breast tissues; DCIS = ductal carcinoma in situ.

**Figure 2 cells-11-02939-f002:**
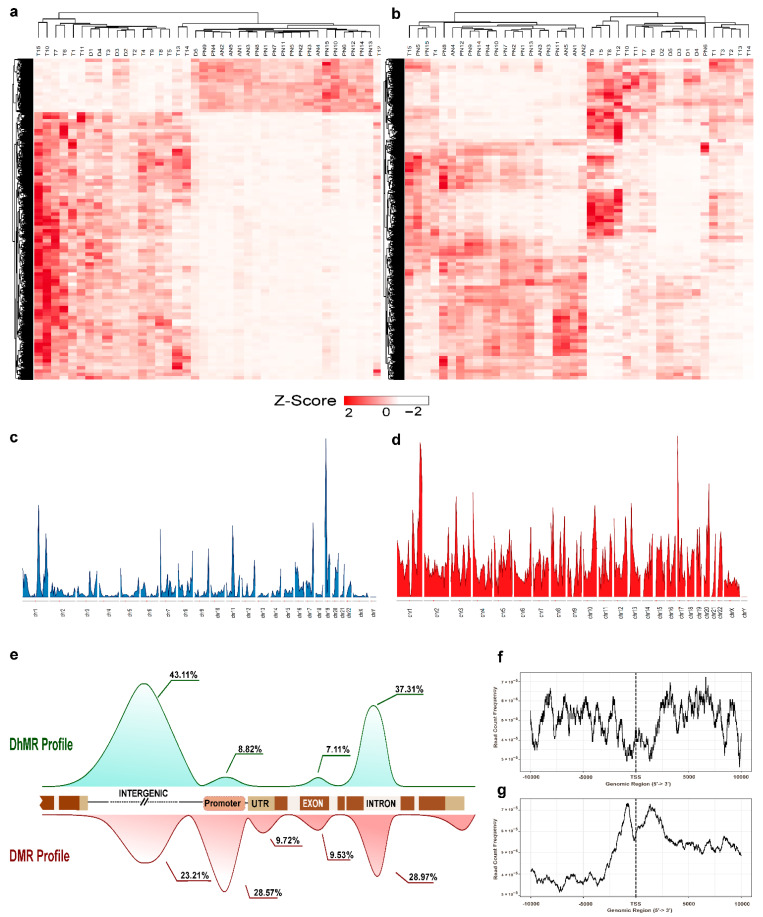
Genome-wide distribution of 5-mC and 5-hmC in breast cancer. (**a**) Heatmap representing DMRs in breast cancer. (**b**) Heatmap showing DhMRs in breast cancer (Z-score ranges from −2 (white) to +2 (red)). (**c**) Chromosomal distribution of differentially methylated regions (DMRs) in breast cancer. (**d**) Chromosomal distribution of differentially hydroxymethylated regions (DhMRs) in breast cancer. (**e**) Genomic features of DMRs and DhMRs in breast cancer. (**f**) Relative peak count frequency of DhMR from Transcription Start Sites (TSSs). (**g**) Relative peak count frequency of DMR from TSSs.

**Figure 3 cells-11-02939-f003:**
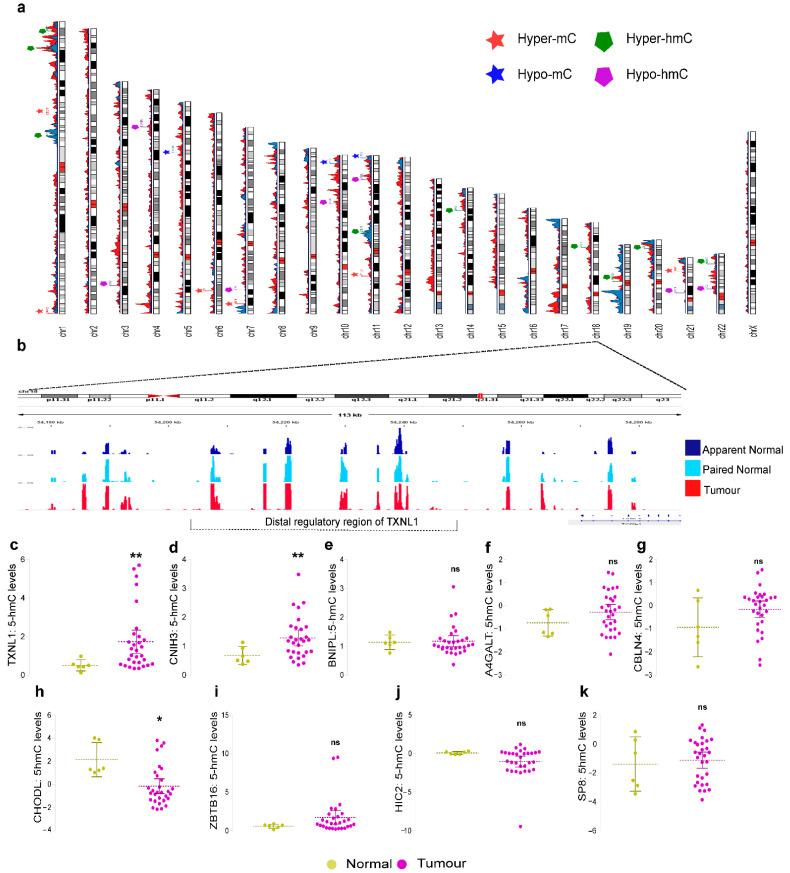
Validation of 5-hmC specific loci in breast tumour and normal samples. (**a**) Ideogram representing DMRs (blue) and DhMRs (red) across all chromosomes and the candidate loci of hyper-mC, hypo-mC, hyper-hmC, and hypo-hmC groups. (**b**) Integrative genome viewer representing the gain of 5-hmC in the distal regulatory region of TXNL1 in tumour, paired normal and apparent normal samples. (**c**) Validation of 5-hmC levels of TXNL1 (**d**) Validation of 5-hmC levels of CNIH3 (**e**) Validation of 5-hmC levels of BNIPL. (**f**) Validation of 5-hmC levels of A4GALT. (**g**) Validation of 5-hmC levels of CBLN4. (**h**) Validation of 5-hmC levels of CHODL. (**i**) Validation of 5-hmC levels of ZBTB16. (**j**) Validation of 5-hmC levels of HIC2. (**k**) Validation of 5-hmC levels of SP8. The *p*-value of <0.05 was considered statistically significant (** *p* < 0.001, and * *p* < 0.05).

**Figure 4 cells-11-02939-f004:**
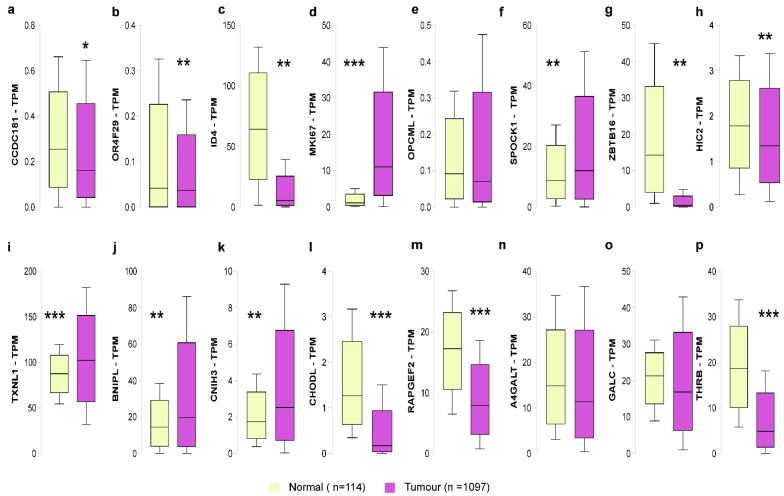
Gene expression analysis of 5-mC and 5-hmC specific loci in breast tumour and normal samples. (**a**–**f**) Gene expression analysis of hyper- and hypo-mC candidates. (**g**–**p**); Gene expression analysis of hyper- and hypo-hmC candidates using UALCAN webtool (*** *p* < 0.0001; ** *p* < 0.001, and * *p* < 0.05).

**Table 1 cells-11-02939-t001:** Loci-specific candidates of 5-mC and 5-hmC in breast cancer.

Hyper-hmC Regions				
Gene Symbol	Entrez Gene Name	Annotated Regions	Log2-Fold Change	*p*-Value
*CBLN4*	Cerebellin 4 precursor	Distal intergenic	1.500	0.00075
*BNIPL*	BCL2 interacting protein-like	Promoter	1.563	0.00108
*CNIH3*	Cornichon family AMPA receptor auxiliary protein 3	Intron	1.542	0.00187
*TXNL1*	Thioredoxin like -1	Distal intergenic	1.720	0.00217
*A4GALT*	alpha 1,4-galactosyltransferase (P blood group)	Promoter	1.520	0.00848
*GALC*	Galactosylceramidase	Distal intergenic	4.915	0.0073
**Hypo-hmC Regions**				
**Gene Symbol**	**Entrez Gene Name**	**Annotated Regions**	**Log2-Fold Change**	***p*-Value**
*SP8*	Sp8 transcription factor	Distal intergenic	−1.312	5.3 × 10^−5^
*CHODL*	Chondrolectin	Intron	−1.570	0.00015
*HIC2*	HIC ZBTB transcriptional repressor 2	Intron	−1.502	0.00355
*RAPGEF2*	Rap guanine nucleotide exchange factor 2 [Homo sapiens (human)]	Distal intergenic	−1.569	0.00587
*ZBTB16*	Zinc finger and BTB domain containing 16	Intron	−1.270	0.00753
**Hypermethylated Regions**				
**Gene Symbol**	**Entrez Gene Name**	**Annotated Regions**	**Log2-Fold Change**	***p*-Value**
*NXPH1*	Neurexophilin 1	Intron	2.810	7.1 × 10^−36^
*SIM2*	SIM bHLH transcription factor 2	Promoter	3.050	2.6 × 10^−31^
*WT1-AS*	WT1 antisense RNA	Promoter	2.792	1.0 × 10^−25^
*CCDC181*	Coiled-coil domain containing 181	Promoter	3.864	3.1 × 10^−14^
*ID4*	Inhibitor of DNA binding 4, HLH protein	Promoter	2.789	3.1 × 10^−12^
*OR4F29*	Olfactory receptor family 4 subfamily F member 29	Distal intergenic	3.001	0.00014
**Hypomethylated Regions**				
**Gene Symbol**	**Entrez Gene Name**	**Annotated Regions**	**Log2-Fold Change**	***p*-Value**
*MKI67*	Marker of proliferation Ki-67	Distal intergenic	−1.266	5.9 × 10^−11^
*SPOCK1*	SPARC (osteonectin), cwcv and kazal-like domains proteoglycan 1	Intron	−1.413	3.4 × 10^−6^
*OPCML*	Opioid binding protein/cell adhesion molecule like	Intron	−1.627	1.9 × 10^−6^

## Data Availability

Raw sequencing data are available in Sequence Read Archive Hosted by National Centre for Biotechnology Information (NCBI) search database with accession number PRJNA769519.

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
