# Peer review of "Locus-Specific Enrichment Analysis of 5-Hydroxymethylcytosine Reveals Novel Genes Associated with Breast Carcinogenesis"

_cells, 2022, doi:10.3390/cells11192939_

Round 1
Reviewer 1 Report
The authors proposed a comprehensive analysis for the epigenomic instability substantiated by the loss of 5-mC and 5-hmC in breast tumors. The study shows the downregulation of TET genes in the tumor samples. The statistical methods are designed properly. It contains differentially expression using Deseq2, pathway enrichment using GO analysis, pathway analysis, and survival analysis was carried by UAL-CAN.
The manuscript is well-presented, however, I have minor changes/suggestions:
- some punctuations e.g. 4841 => 4,841.
- The figures are a bit fuzzy. The authors may generate the figures in vector based format, or higher resolution format.
Reviewer 2 Report
Many researchers agree that an imbalance in DNA methylation is a hallmark of epigenetic changes in cancer. The conversion of 5-methylcytosine (5-mC) to 5-hydroxymethylcytosine (5-hmC) causing imbalance leads to aberrant gene expression. The exact functional role of 5-hydroxymethylcytosine in breast cancer remains unclear. The authors of this article described the landscape of 5-mC and 5-hmC and their association with the development of breast cancer. A discernible global loss of 5-hmC has been shown to occur in localized and invasive types of breast cancer, which is highly correlated with TET expression. Genome-wide analysis revealed a unique signature of 5-mC and 5-hmC in breast cancer. The authors identified 2 new 5-hmC candidates, such as TXNL1 and CNIH3, with a putative pro-oncogenic role in the development of breast cancer. I liked the article, the design of the study raises no questions, the material is structured and consistently presented. From minor remarks - the captions in the figures are too small, in places it is impossible to read. Table 1 shows log2 values with 9 decimal places, why? In the p-values, the significant digits must also be ordered.
